# Emerging Insights into Keratin 16 Expression during Metastatic Progression of Breast Cancer

**DOI:** 10.3390/cancers13153869

**Published:** 2021-07-31

**Authors:** Maha Elazezy, Sandra Schwentesius, Luisa Stegat, Harriet Wikman, Stefan Werner, Wael Y. Mansour, Antonio Virgilio Failla, Sven Peine, Volkmar Müller, Jean Paul Thiery, Majid Ebrahimi Warkiani, Klaus Pantel, Simon A. Joosse

**Affiliations:** 1Department of Tumor Biology, University Medical Center Hamburg-Eppendorf, 20246 Hamburg, Germany; m.elazezy@uke.de (M.E.); s.schwentesius@uke.de (S.S.); luisa@leifheit-stegat.de (L.S.); h.wikman@uke.de (H.W.); st.werner@uke.de (S.W.); pantel@uke.de (K.P.); 2Department of Radiotherapy and Radiation Oncology, University Medical Center Hamburg-Eppendorf, 20246 Hamburg, Germany; wmansour@uke.de; 3UKE Microscopy Imaging Facility (UMIF), University Medical Center Hamburg-Eppendorf, 20246 Hamburg, Germany; a.failla@uke.de; 4Department of Transfusion Medicine, University Medical Center Hamburg-Eppendorf, 20246 Hamburg, Germany; s.peine@uke.de; 5Department of Gynecology, University Medical Center Hamburg-Eppendorf, 20246 Hamburg, Germany; v.mueller@uke.de; 6Bioland Laboratory, Guangzhou Regenerative Medicine and Health Guangdong Laboratory, Guangzhou 510320, China; tjp@nus.edu.sg; 7School of Biomedical Engineering, University of Technology Sydney, Sydney 2007, Australia; majid.warkiani@uts.edu.au

**Keywords:** circulating tumor cells (CTCs), keratin 16 (*KRT16*), epithelial to mesenchymal transition (EMT)

## Abstract

**Simple Summary:**

The mechanisms leading to tumor metastasis remain poorly understood, and therefore, phenotyping of circulating tumor cells from cancer patients may contribute to translating these mechanisms. In in silico analysis, high expression of keratin 16 was associated with higher tumor aggressiveness. According to our results, keratin 16 is a metastasis-associated protein that promotes EMT and acts as a positive regulator of cellular motility by reorganizing the actin cytoskeleton, which is the driving force behind disrupting intercellular adhesion and directional migration. In metastatic breast cancer patients, circulating tumor cells expressing keratin 16 were associated with shorter relapse-free survival. This is an important issue for future research to determine the exact function of keratin 16 in tumor dissemination and metastasis development by analyzing keratin 16 status in disseminating tumor cells. Furthermore, gaining a better knowledge of keratin 16’s biology would give crucial mechanistic insights that might lead to a unique treatment option.

**Abstract:**

Keratins are the main identification markers of circulating tumor cells (CTCs); however, whether their deregulation is associated with the metastatic process is largely unknown. Previously we have shown by in silico analysis that keratin 16 (*KRT16*) mRNA upregulation might be associated with more aggressive cancer. Therefore, in this study, we investigated the biological role and the clinical relevance of K16 in metastatic breast cancer. By performing RT-qPCR, western blot, and immunocytochemistry, we investigated the expression patterns of K16 in metastatic breast cancer cell lines and evaluated the clinical relevance of K16 expression in CTCs of 20 metastatic breast cancer patients. High K16 protein expression was associated with an intermediate mesenchymal phenotype. Functional studies showed that K16 has a regulatory effect on EMT and overexpression of K16 significantly enhanced cell motility (*p* < 0.001). In metastatic breast cancer patients, 64.7% of the detected CTCs expressed K16, which was associated with shorter relapse-free survival (*p* = 0.0042). Our findings imply that K16 is a metastasis-associated protein that promotes EMT and acts as a positive regulator of cellular motility. Furthermore, determining K16 status in CTCs provides prognostic information that helps to identify patients whose tumors are more prone to metastasize.

## 1. Introduction

Breast cancer is a heterogeneous disease encompassing different molecular subtypes that are identified based on their hormone status and/or gene expression patterns [1]. Long-term survival of breast cancer patients largely depends on when the primary tumor and especially the metastases are detected [2]. Epithelial to mesenchymal transition (EMT) is thought to play an essential role in initiating cancer dissemination and metastasis. During this process, intercellular adhesive complexes, such as E-cadherin-based adherens junctions, are downregulated, leading to a mesenchymal-like phenotype [3,4]. The reverse process, i.e., mesenchymal to epithelial transition (MET), plays a critical role in metastatic tumor formation [5]. High plasticity of carcinoma cells enables them to undergo a dynamic and reversible transition between the epithelial and mesenchymal-like phenotype [6].

An important aspect of EMT is the reorganization of the cytoskeleton, including changes in intermediate filaments, which may contribute to the induction of cell motility. Keratins are intermediate filament proteins routinely used for cancer diagnostics [3,7]. Keratins are mainly present in epithelial cells anchored to desmosomes, hemidesmosomes, and the nuclear membrane. Keratins contribute to the control of cell shape and rigidity [8], as well as regulating intracellular signaling pathways [9]. The keratin 16 (*KRT16*) gene is located at chromosome 17q21.2, encoding the type I cytoskeletal 16 protein K16 [9]. Previous studies have shown that K16 expression influences keratinocyte organization, which contributes to the changes in the morphology of epithelial cells and directly impacts adhesion, differentiation, and migration of cells during wound-healing [10,11,12].

Little is known about the deregulation of K16 in cancer and metastasis. Through in silico analysis, a positive correlation between *KRT16* gene expression and shorter relapse-free survival was shown in two large breast cancer patients’ data sets [3]. These data indicate that *KRT16* expression is associated with higher tumor aggressiveness and shorter relapse-free survival. To further elucidate the role of K16 in cancer progression and metastasis, this study set out to investigate the biological role of K16 in metastatic breast cancer cell lines and evaluate the clinical relevance of K16 in metastatic breast cancer patients by analyzing the K16 expression in CTC, i.e., the seeds of metastasis.

## 2. Materials and Methods

### 2.1. In Silico Analysis

*KRT1-20*, *CDH1*, and *VIM* gene expression data of 51 breast cancer cell lines were obtained from GEO, accession number GSE69017 [13]. A hierarchical cluster analysis was performed on the gene expression data to compare expression levels of *KRT1-20* in the different breast cancer molecular subtypes and to evaluate the epithelial- and mesenchymal-like phenotype based on *CDH1* and *VIM*. The dataset was normalized to the mean value of each probe set.

### 2.2. Cell Culture

Ten human breast cancer cell lines (MDA-MB-468, MDA-MB-231, BT549, HS-578T, MCF7, T47D, MDA-MB-361, BT474, SKBR3, GI-101A), one normal-like breast epithelial cell line (MCF-10A), and one skin squamous carcinoma cell line (A431) were brought into a culture. All cell lines were obtained from ATCC. The cells were cultured in either DMEM media (catalog no. P04-03600, Aidenbach, Germany) or RPMI 1640 media (catalog no. P04-17500, Aidenbach, German) and incubated at 37 °C and 5% CO_2_ or 10% CO_2_, according to ATCC’s instructions. Both media were supplemented with 10% fetal bovine serum (FBS) (Gibco—Life Technologies), 1% L-glutamine (catalog no. 25030-024, Gibco—Life Technologies), and 1% penicillin/streptomycin (catalog no. 15140-122, Gibco—Life Technologies). All cells were grown in a 25 cm^2^ flask until confluence was reached. Cells were washed with DPBS (catalog no. 14190-094, Gibco, Life Technologies) and harvested using trypsin/EDTA (catalog no. 25200-072; Thermo Fisher Scientific). A test for mycoplasma was regularly performed in all cultures to detect and prevent any potential mycoplasma contamination [14].

### 2.3. EMT Induction Assay

EMT was induced using StemXVivo EMT Inducing Media Supplement (catalog no. CCM017; R & D Systems, Wiesbaden-Nordenstadt, Germany). This media includes a cocktail of E-cadherin, SFRP-1, and DKK-1 blocking antibodies and WNT-5 and TGF-β1 recombinant proteins. MCF7 cells were seeded in standard culture media containing 1× StemXVivo EMT Inducing Media Supplement according to the manufacturer’s instructions [15,16]. Different culture conditions were tested through EMT treatment, such as hypoxia (1% O_2_) and starvation (0.5% FCS) compared to standard conditions (21% O_2_, 10% FCS) at different time-points of 24 h, 72 h, and 120 h. Control cells were seeded and put through the same conditions as EMT-treated cells. The experiment was processed in duplicate. Characterization of EMT-induced cells was performed by western blot and RT-qPCR.

### 2.4. KRT16 Overexpression

Keratin 16 plasmid DNA (catalog no. OHu24939D; GenScript) and pcDNA3.1+/c-(K)-DYK vector were used for transfection into MCF7 cells to overexpress *KRT16* and as a vector control, respectively. ORFs cloned in the pcDNA3.1+/C-(K) DYK vector were expressed in MCF7 cells as a tagged protein with a C-terminal DYKDDDDK tag. First, cells were seeded in 6-well plates (9 × 10^5^ cells/well) in a standard medium containing 10% FBS, the day prior to transfection. Then cells were transfected with plasmid constructs (final concentration 2.5 µg) using Lipofectamine 3000 (catalog no. L3000008; Invitrogen) and Opti-MEM medium (Thermo Fisher Scientific), following the manufacturer’s protocol [17]. The experiment was processed in triplicate. Keratin 16 overexpression was assessed by western blot using a DYKDDDDK Tag monoclonal anti-mouse clone [5A8E5] (catalog no. A00187; GenScript) and RT-qPCR. Transfection efficiency was assessed by immunofluorescence staining. After 24 h of transfection, cells were fixed in 4% paraformaldehyde (PFA) for 15 min on a culture slide and washed with PBS two times. Next, the cells were incubated with 0.2% Triton X-100 for 10 min and washed with PBS three times. Then, the cells were incubated with 1% BSA at room temperature for 30 min, followed by incubation with the primary antibody against DYKDDDDK Tag overnight at 4 °C. On the second day, cells were washed with PBS and incubated with the secondary antibody goat anti-mouse IgG labeled with Alexa Fluor 488 (catalog no. A28175; Life technologies, 1:200) at room temperature for 90 min. After being washed with PBS, the cells were incubated for 1 min with DAPI (Janssen Diagnostics, 1:5000). The cells were examined by fluorescence microscopy (ZEISS Axio Observer). Pictures were taken for four fields and cells were counted by the Cell counter program [18]. The experiment was performed in duplicate. The results are expressed as the average percentage number of positive cells within a transfected cell population relative to the total number of cells.

### 2.5. KRT16 Knockdown

Transfection using a smart pool of *KRT16* interfering RNA (siRNA) duplexes, *KRT16*siRNA1 (GGAGAUGCUUGCUCUGAGA); *KRT16*siRNA2 (GGCCAGAGCUCCUAGAACU); *KRT16*siRNA3 (GGAACAAGAUCAUUGCGGC); *KRT16*siRNA4 (GCGGAGAUGUGAACGUGGA) (catalog no. L-017550-02-0005; Dharmacon), and Lipofectamine™ RNAiMAX Transfection Reagent (catalog no. 13778075; Thermofisher) was performed on MDA-MB-468 cells, according to the manufacturer’s protocol [19]. ON-TARGETplus Non-targeting Pool (catalog no. D-001810-10-05; Dharmacon) was used as a control. The experiment was processed in duplicate, with a final oligonucleotide concentration of 20 nM. *KRT16* knockdown efficiency was assessed by RT-qPCR, and western blot and cells were used for further experiments 48 h after transfection.

### 2.6. Protein Level Assessment

Protein levels of the cells were measured after treatment with StemXVivo EMT Inducing Media, *KRT16* knockdown, and *KRT16* overexpression as follows: the cells were scraped in PBS and centrifuged at 1500× *g* for 2 min, the supernatant was removed, and the cell pellet was resuspended in RIPA lysis buffer containing the Protease Inhibitor Cocktail (Thermo Fisher Scientific). After incubation on ice for 30 min, cells were homogenized by ultrasonic treatment for 5 s and centrifuged at 16100× *g* for 15 min at 4 °C. The protein concentration was determined by a BSA protein assay kit (catalog no. 23227; Thermo Scientific), according to the manufacturer’s instructions [20].

Protein extracts were loaded in 1x SDS buffer and denatured at 95 °C for 5 min. The denatured proteins were separated by SDS-PAGE using 10% polyacrylamide gels and blotted onto PVDF membrane. Detection of proteins was performed by incubation with the following specific antibodies: keratin 16 monoclonal anti-mouse clone [Ag11240] (catalog no. 66802-1-Ig; Proteintech), E-cadherin monoclonal anti-rabbit clone [EP700Y] (catalog no. ab40772; Abcam), vimentin monoclonal anti-mouse clone [RV202] (catalog no. 550513; BD Pharmingen™), SNAI2 monoclonal anti-mouse clone [A7] (catalog no. sc-166476; Santa Cruz Biotechnology), and HSC70 monoclonal anti-mouse clone [B-6] (catalog no. sc-7298; Santa Cruz Biotechnology). Protein bands were determined using SignalFire™Plus ECL reagent (Cell Signaling Technology, Danvers, USA), X-ray films (CEA, Hamburg, Germany), and LI-COR C-DiGit Chemiluminescence Western Blot Scanner, according to the manufacturers’ instructions [21].

### 2.7. Quantitative Transcript Analysis

Relative RNA expression levels were measured by RT-qPCR using an equivalent of 15 ng of total RNA, isolated using the NucleoSpin RNA kit (catalog no. 740955.50, Macherey Nagel), according to manufacturer’s instructions [22]. The RNA was reverse transcribed, performing the First Strand cDNA Synthesis Kit (Thermo Scientific) according to the manufacturer’s instructions. Primers and product length for each gene are described in Table 1. AccuPower^®^ 2X GreenStar™ master mix solution (catalog no. K-6253, Bioneer) RT-qPCR was used in 10 μL reaction volumes containing 10 pmol of each primer, 15 ng cDNA, and 1× master mix (Tris-HCl, 60 mM KCl, 1.5 mM MgCl_2_, SYBR Green I Dye, Hotstart DNA polymerase (1U), dNTP mixture (each 250 µM)). The PCR conditions were as follows: initial denaturation at 95 °C for 5 min, followed by 40 amplification cycles of 95 °C for 15 s and 61.4 °C for 30 s, and a melting curve of 65.0 °C to 95.0 °C, with increments of 0.5 °C every 5 s. The RT-qPCR reactions were run in duplicate using a CFX96 Touch™ Real-Time PCR Detection System. Data were analyzed by applying the ΔΔCT calculations [23], using the GAPDH expression for normalization to calculate mRNA expressed as fold changes 2^−(∆∆Ct)^.

### 2.8. Migration Assay

Two migration assays were performed in this study, the Wound-healing and transwell (Boyden chamber) assays, to ensure the viability of modulated cells throughout the migration process. For the wound-healing assay, 1.2 × 10^6^ MDA-MB-468 cells were plated in serum-free DMEM media in a 6-well plate and incubated at 37 °C. After 24 h, a scratch was made, and pictures were taken under the microscope at different time-points. The experiment was processed in duplicate. The images were analyzed by ImageJ software (MRI_Wound_Healing_Tool.ijm) [24]. For the transwell migration assay, 1 × 10^5^ MCF7 cells were cultured in serum-free DMEM media in the upper chambers, with 8.0 μm pores of BD Cell Culture (BD Falcon). DMEM containing 10% FCS was used as a chemoattractant in the lower chamber. Plates were incubated under standard conditions, and migration could proceed for 24 h. Non-migrated cells in the upper chambers were removed with cotton swabs, and the remaining cells were fixed in 4% PFA and stained with crystal violet. Using a converted microscope (ZEISS Axio Observer), pictures were taken of four fields and counted by the Cell counter program [18]. The experiment was performed in triplicate, and the results are expressed as the average number of cells with standard deviation.

### 2.9. Proliferation Assay

Cell proliferation was assessed after *KRT16* knockdown and *KRT16* overexpression. The cells were seeded and incubated at 37 °C with 5% CO_2_. Monitoring the cell density and viability over time was measured in triplicate with the Vi-CELL™ XR 2.04 (Beckman Coulter), which depended on performing the Trypan Blue Dye Exclusion method [25]. The experiment was performed in triplicate, and the results are represented as the average number of cells.

### 2.10. Immunocytochemical Staining

To visualize the actin filaments and a cell–cell adhesion protein (E-cadherin) in K16-expressing cells, staining was performed using phalloidin Alexa Fluor 555 (catalog no. A34055; Life Technologies, 1:00) and E-cadherin monoclonal anti-rabbit clone [EP700Y] (catalog no. ab40772; Abcam, 1:500) as follows: 30,000 cells were seeded onto a chamber slide and incubated at the standard culture condition for 48 h. The cells were fixed with 4% PFA for 10 min and incubated with 0.1% Triton X-100 for 10 min, 10% AB-serum at room temperature for 20 min, primary antibody against E-cadherin overnight at 4 °C, and secondary antibody goat anti-rabbit IgG labeled with Alexa Fluor 488 (catalog no. A1008; Life technologies, 1:200) at room temperature for 60 min. Cells were washed with PBS, incubated with phalloidin and DAPI (1:1000) for 60 min, and examined by confocal microscopy (Leica TCS SP8). Using the Bitlplane Imaris software, the cell thickness was estimated by counting the number of image planes required to cover the entire cell volume. Actin filament length was determined by calculating the distance between the extreme points of the detected filament structures. Movies were produced after 3D reconstruction by the mean of image interpolation of confocal images stacks, voxel (80 × 80 × 300) nm.

### 2.11. Blood Collection and Processing

Twenty metastatic breast cancer patients were included into the study after giving written informed consent (local ethical committee approval number: PV5392). Patients were treated at the Department of Gynecologic Oncology, University Medical Center Hamburg-Eppendorf and received therapy according to international guidelines. The selection criteria were women with metastatic breast cancer, independent of the tumor’s hormone status. Blood was taken upon a progression of the disease. Peripheral blood samples were collected into EDTA-containing tubes, kept at room temperature, and processed within 1 h. The density gradient Ficoll (catalog no. 17-1440-03; GE Healthcare) was used for mononuclear cell enrichment as before [26].

### 2.12. Enrichment of CTCs using a Spiral Microfluidic Chip

A spiral microfluidic chip was designed to isolate CTCs according to size and density [27], in which the larger and denser particles (i.e., CTCs) focused and aligned near the inner wall, while the small particles occupied the lateral positions near the outer wall. One inlet and two outlet tubes were connected to a spiral chip. The inlet tubing was connected to a syringe pump (catalog no. 78-8110; Kd. Scientific), and the outlet tubing was connected to two sterile 15 mL collection BD Falcon tubes. An initial washing was performed before sample processing using 5 mL of 5% NaClO, 5 mL H_2_O, and finally 5 mL sterile 1× PBS at a flow rate of 1 mL/min. The sample was transferred into a syringe and pumped through the spiral chip at a flow rate of 1.7 mL/min. The enriched CTC fraction was collected on a microscope slide in a cytospin funnel and spun down at 190× *g* for 5 min.

### 2.13. CTC Immunophenotyping

K16 staining was established using 7.5 mL blood samples from anonymous healthy donors spiked with tumor cell line cells. The blood was processed as mentioned above, and the mononucleated cell layer was collected and spiked with MCF7 cells (K16−/C11+) as the K16 negative control and A431 cells (K16+/C11−) or MDA-MB-468 (K16+/C11+) as K16 positive control. The spiked samples were spun on a microscope slide. Immunofluorescence staining was used to identify the enriched CTCs and spiked tumor cell line cells. Briefly, cells were fixed in 4% PFA for 10 min on a microscope slide and washed with PBS three times. Next, the cells were incubated with 10% AB-serum at room temperature for 1 h, followed by incubation with the primary antibody against keratin 16 (catalog no. 66802-1-Ig; Proteintech, 1:500) overnight at 4 °C. On the second day, cells were washed with PBS and were incubated with the secondary antibody rabbit anti-mouse IgG labeled with Alexa Fluor 546 (catalog no. A11060; Life technologies, 1:200) at room temperature for 1 h. After being washed with PBS, the cells were incubated for 1 h with DAPI (1:1000), CD45 Alexa Fluor 647 (catalog no. 130-110-633; MACS, 1:200), and C11 Alexa Fluor 488 (catalog no. ab187773; Abcam, 1:300) in order to detect keratins (K4/5/6/8/10/13/18). After staining, the cells were washed with PBS, covered with a coverslip, and examined by fluorescence microscopy (ZEISS Axio Observer). CD45−/K16+/C11+ cells with an intact nucleus were interpreted as CTCs.

### 2.14. Statistical Analyses

Statistical analyses were performed using R (R Foundation for Statistical Computing, version 4.0.1) and In-Silico Online, version 2.3.0 [28], and graphs were generated using GraphPad Prism (GraphPad Software, San Diego, CA, USA). Mean values were given with standard deviations. Statistical significance was defined as *p* < 0.05. Linear regression was used to determine the rate of migrated cells over time. Relapse-free survival (RFS) was determined using a Kaplan–Meier curve and log-rank test. The primary endpoint of RFS was defined as the time in months from primary diagnosis until the first progression. The clinical variables (estrogen receptor (ER) status, progesterone receptor (PR) status, ERBB2 status, T-stage, and age) were compared to the K16-positive CTCs of metastatic breast cancer patients. A power analysis for sample size was performed by PASS, version 20.0.2.

## 3. Results

### 3.1. Keratin 16 Is Overexpressed in the Basal-Like Breast Cancer Subtype

In in silico analysis, we investigated the expression of 20 keratins in 54 breast cancer cell lines (GSE69017, Appendix A) and could show that *KRT16* was upregulated in 24% (13/54) of breast cancer cell lines of predominantly the basal-like (53.8%), ERBB2 enriched (23%), claudin-low (15%), and normal-like (7.7%) subtypes that mainly overexpress *CDH1* and *VIM*, whereas *KRT16* was downregulated in cell-lines of the luminal A and luminal B molecular subtypes. Furthermore, we investigated the correlation between mRNA expression levels of *KRT16* and the EMT-associated genes *CDH1* and *VIM*. A significant positive correlation (R^2^ = 0.643, *p* = 0.018, Pearson’s r) was observed between *CDH1* and *KRT16*. In order to extend these data at the protein level, we assessed the protein expression of K16 in different breast cancer cell lines presenting various breast cancer subtypes using western blot analysis. In addition to K16, the expressions of E-cadherin (CDH1) and Vimentin (VIM) were determined to assess the degree of epithelial- and mesenchymal-like phenotype, respectively. Western blot analysis revealed that K16 protein was more abundant in carcinoma cells of the basal-like A and normal-like subtypes that also express CDH1 and VIM, while K16 expression was completely absent in cell lines of the luminal A and B subtypes that express CDH1 but not VIM and the ERBB2-overexpressing subtype that does not express CDH1 and VIM (Figure 1A,B). A significant correlation between mRNA and protein levels in the expression of CDH1 (R^2^ = 0.851, *p* = 0.0001), VIM (R^2^ = 0.653, *p* = 0.0047), and K16 (R^2^ = 0.773, *p* = 0.0008) were observed (Figure 1C). Taken together, these data reveal a correlation between K16 expression and mesenchymal-like phenotype.

### 3.2. EMT Induction Leads to Overexpression of K16

Next, we sought to address whether the induction of the EMT process leads to K16 upregulation. Therefore, we investigated changes in K16 expression during EMT induction in the MCF7 cell line, which normally does not express K16 (Figure 1A). Microscope imaging of EMT-induced MCF7 cells showed substantial changes in their morphology after treatment with the EMT-inducing media supplement. The EMT-induced cells showed a mesenchymal-like, spindle-shaped morphology, losing all intercellular contacts, whereas the untreated cells showed a typical epithelial-like morphology with extended cell–cell contacts (Figure 2A). In order to assess EMT induction efficiency, the mRNA expression levels of the MET/EMT related genes *CDH1*, *KRT8*, *KRT18*, *KRT19*, *CD24*, *VIM*, *CDH2*, *SNAl2*, *CD44*, *ZEB1*, *ZEB2*, *SNAI1*, *TWIST1*, *WNT5A*, and *NOTCH1* were examined at various EMT culture conditions and different time-points (Figure 2B). Mesenchymal markers *VIM*, *CDH2*, *SNAI2*, and *CD44* were significantly (*p <* 0.0001) upregulated in the EMT-induced cells as compared to untreated controls, in particular, the cells that were under standard conditions (10% FCS, 21% O_2_) and starvation conditions (0.5% FCS, 21% O_2_) (Figure 2B). These cells showed slight changes in the expression of epithelial markers, such as *CDH1* and *KRT8*. We further observed that EMT-induced cells could continue to transition after five days (Figure 2B).

Interestingly, K16 was upregulated in EMT-induced cells under all conditions in which cells were exposed at all time points (Figure 2B). Immunoblot analysis showed overexpression of K16 and SNAI2 (Figure 2C,E,F) with decreased expression of CDH1 compared to untreated cells (Figure 2C,D), which indicates the disruption in adhesion junction formation and loss of epithelial properties, whereas no change in VIM expression was detected (Figure 2C). Overall, EMT promotes the expression of K16 which highlights a crucial role of K16 in EMT execution.

### 3.3. Overexpression of KRT16 Leads to EMT Induction

To further investigate the association between K16 and the EMT, *KRT16* was overexpressed in the K16-negative MCF7 cells (Figure 1A). MCF7 cells were transfected with a *KRT16* coding sequence containing plasmid or non-target plasmid as a control, resulting in transfection efficiency of 81.4% of the total transfected cell population (Appendix A and Appendix A). Transfection of *KRT16* increased the mean fold-change of mRNA gene expression levels by 30,000-fold in the treated MCF7 cells, compared to the non-target treated control cells (Figure 3A). Furthermore, K16 protein expression was strongly induced in the treated MCF7 cells, whereas no expression of K16 could be detected in the non-target treated control cells (Figure 3B,C). The treated MCF7 showed a spindle-shaped morphology with long and thin stress fibers, characteristic of epithelial-to-mesenchymal transition, whereas non-target treated and untreated control cells were regular in morphology (Figure 3E and Appendix A). In line with this, mRNA analysis revealed overexpression of the mesenchymal specific genes *VIM*, *CDH2*, *SNAI2*, *ZEB1*, *ZEB2*, *TWIST1*, and *WNT5A* in *KRT16* induced cells (*p* = 0.0001, Two-sample Wilcoxon rank test; Figure 3F), but no changes in the expression levels of *CDH1*, *KRT8*, *KRT18*, *KRT19* and *CD24*. Furthermore, no changes were detected in VIM and CDH1 protein expression levels (*p* = 0.430; Figure 3B,D) upon *KRT16* induction. Immunocytochemical staining showed abundant expression of E-cadherin localized in the cytoplasm and weakening cell–cell adhesion in K16-induced MCF7 cells, whereas untreated MCF7 cells demonstrated stable adherens junctions at the cell edges (Figure 3G). Moreover, overexpression of K16 resulted in a reorganization of actin microfilaments to form long, parallel, thin stress fibers compared to untreated MCF7 cells (Figure 3G and Appendix A). A highly significant increase (*p* = 0.0027, Welch’s two sample *t*-test) in the length of actin microfilaments was detected with a reduction in the cell thickness (*p* = 0.0162, Welch’s two sample *t*-test) of MCF7 cells that induced K16 compared to control cells (Figure 3H,I).

The phenotypic changes induced by *KRT16* overexpression were further investigated by using the Boyden chamber assay for migration analysis (Figure 3J). Compared to the non-target treated cells, MCF7 cells overexpressing *KRT16* had a significantly higher mean migration rate of 1.88 times (95% CI: 1.47–2.42; *p <* 0.001, negative-binomial generalized linear mixed-effects model; Figure 3K). The equal mean proliferation rates of 79.4 and 78.3 of the target and empty-vector-control treated cells, respectively, indicate that the proliferation rate was not modified by *KRT16* overexpression and that the increased migration was not the consequence of a higher proliferation rate (*p* = 0.8514, Welch’s two sample *t*-test; Figure 3L). Hence, we could show that overexpression of *KRT16* contributes to EMT and to a more aggressive phenotype.

### 3.4. K16 Knockdown Changes the Mesenchymal Phenotype

Next, we sought to investigate whether K16 expression contributes to EMT-associated cell (de)differentiation. To that end, *KRT16* was knocked down in MDA-MB-468 cells using siRNA directed against *KRT16* mRNA. MDA-MB-468 is a basal-like cell line expressing K16, but it also contains cells having an epithelial-like phenotype (Figure 1A). *KRT16* was successfully depleted, leading to a mean downregulation of *KRT16* mRNA of five times compared to control cells (*p* < 0.0001, Welch’s *t*-test; Figure 4A). Because of gene silencing by siRNA, protein K16 levels were downregulated to 0.18 as compared to the control cells (Figure 4B,C). In addition, CDH1 and VIM expression were lowered by 0.56 and 0.35 in K16-depleted cells, respectively, compared to untreated cells (Figure 4B,D,E), indicating a less mesenchymal phenotype upon K16-knockdown. Furthermore, the mRNA expression of selected mesenchymal-associated genes (*VIM*, *CDH2*, *SNAI2*, *CD44*, *TWIST1*, *NOTCH1*, *ZEB1*, *ZEB2*, *SNAI1*, and *WNT5A*) were significantly downregulated (*p* = 0.0135, Two-sample Wilcoxon rank test), whereas the expression of selected epithelial-associated genes (*CDH1*, *KRT8*, *KRT18*, *KRT19*, and *CD24*) stayed unaltered (*p* = 0.127, Two-sample Wilcoxon rank test; Figure 4F), indicating that MDA-MB-468 expressed mesenchymal markers to a lesser extent after *KRT16* knockdown.

Because one of the characteristics of breast cancer cells with a mesenchymal-like phenotype is increased migration properties, we assessed whether knockdown of *KRT16* impacts cell migration. Using the wound-healing assay, it could be observed that *KRT16* knockdown led to a mean 70% slower migration (Figure 4G,H; *p* = 0.0032, ANCOVA test), indicating that the depletion of K16 protein expression in the MDA-MB-468 cell line impaired cell migration. Equal means of proliferation rates in the siRNA treated and control cell lines confirmed that the migration was not the consequence of a higher proliferation rate (*p =* 0.9714; Wilcoxon rank-sum test, Figure 4I). Taken together, these results indicate that loss of K16 directly influences the capacity of cells to migrate, which might be the consequence of loss of mesenchymal properties.

### 3.5. K16 Expression in CTCs Correlates with a Worse Survival

To evaluate the clinical relevance of the expression of K16 in CTCs, an immunofluorescence staining protocol was established by spiking blood with K16 positive and negative cancer cell lines (Figure 5A). Using the optimal staining conditions, specific detection of tumor cell line cells MDA-MB-468 and A431 was achieved with no background in the K16-negative cell line MCF7.

Next, blood was acquired from 20 metastatic breast cancer patients that were experiencing disease progression. CTCs (keratin+/DAPI+/CD45−) were detected in 19/20 patients, follow-up was not available for one patient (Appendix A). The blood of five patients counted ≤5 CTCs, and 14 patients counted ≥5 CTCs per 7.5 mL blood. A total of 64.7% of detected CTCs were K16+/C11−, and 16.6% were C11+/K16−, whereas 18.7% of detected CTCs were K16+/C11+ (Figure 5B and Appendix A).

The median follow-up time of breast cancer patients was 32 months. Relapse-free survival (RFS) analysis was performed to test the difference in survival between metastatic breast cancer patients diagnosed with K16 expressing CTCs only (i.e., K16+/C11− and/or K16+/C11+) and patients that were diagnosed with at least one K16-negative CTC (i.e., C11+/K16−), patients without detectable CTCs were excluded from this analysis. Patients with K16+ CTCs had shorter relapse-free survival (median: 12.9 months) compared to patients who had CTCs with negative expression of K16 (median: 75 months; *p* = 0.0042; Figure 6).

Both in uni-and multivariable analysis, the presence of K16-positive CTCs was correlated with shorter survival (Table 2). Although the expression of estrogen receptor (ER) was correlated with improved RFS in multivariable analysis, ER-status was not significantly associated with survival in univariable analysis, possibly due to the small number of cases. These results indicate that K16 expression is independently associated with high aggressiveness.

## 4. Discussion

Tumor metastasis is the major cause of cancer-related deaths. Therefore, identifying a potential biomarker that can detect micrometastasis early could be helpful in clinical tumor management. We have previously shown by in silico analysis that K16 mRNA expression upregulation might be associated with a more aggressive course of cancer [3]. Accordingly, we investigated the biological role of K16 in metastatic breast cancer. In this study, K16 was positively correlated to an intermediate mesenchymal phenotype, and this K16 expression was mainly observed in cell lines that had a hybrid phenotype of epithelial and mesenchymal cell features. These results provide further support for the hypothesis that K16 appears to contribute to the hyperplastic epithelial phenotype.

EMT has been recognized as an important mechanism driving cell migration and invasion during tumor growth; however, the process of partial and full reversion of EMT in the stages of metastasis is not well understood yet [29,30]. Although our results indicate a clear association between *KRT16* expression and EMT regulation, further experiments have to be performed to determine whether *KRT16* is upstream or downstream of EMT or whether it can function in a transcription-translation feedback loop. The relative gene and protein expression of epithelial markers such as CDH1 were not changed upon transfection of K16 in MCF7 cells; E-cadherin was relocalized from the cell membrane to the cytoplasm and thereby compromised cell–cell adhesion, as well as acting possibly as transcriptional gene regulator [4,31]. Our findings are similar to the study reported by Huang and colleagues who showed the phenotypic EMT characterization of 43 ovarian cancer cell lines and found that CDH1 expression in the cytoplasm was correlated with an intermediate mesenchymal phenotype [32]. Furthermore, the knockdown of *KRT16* also led to the downregulation of mesenchymal-associated genes but not epithelial-related genes. While the expression level of CDH1 protein was reduced in K16-depleted cells, our results suggest that expression of K16 is correlated with the regulation of the mesenchymal-regulatory genes of EMT, as well as the expression of CDH1 as a crucial epithelial marker. Overall, the functional role of K16 in the EMT appears to be essential to allow epithelial carcinoma cells to undergo multiple morphological and biochemical changes to have higher plasticity and thus be able to migrate. Carcinoma cells could acquire a mesenchymal-like phenotype through EMT before or during intravasation. A reverse mechanism, i.e., MET, may occur at the secondary site following extravasation (Figure 7). A recent study reported evidence that the transcription factor TF-AP2A in EMT-related pathways induced K16 expression in lung adenocarcinoma [33]. This study is consistent with our finding that K16 has a regulatory role in EMT, and thus, more in-depth studies are needed to provide additional information on the role of K16 during EMT and MET.

K16 is one of the intermediate filament members that forms a heterodimer interaction with K6 but is also able to form homodimers [9]. Homomeric K16 formation further displays unique properties that may contribute to morphological changes of epithelial cells during migration [12,34]. In the present study, modifications of the actin microfilament morphology has been observed in MCF7 cells under conditions inducing K16 expression. These changes demonstrate the mechanical effect of K16 in the reorganization of intermediate filaments, which is required for changes in cell shape and locomotion as the first step to induce migration [2]. In this study, K16 was shown to significantly enhance the migration in MCF7 cells by reorganizing the actin cytoskeleton, which is the driving force behind disrupting intercellular adhesion and directional migration. In contrast, K16-depleted cells exhibited impaired cell migration. In accordance with the present results, previous studies on wound-healing have demonstrated a crucial role of K16, along with other partners such as K6 and K17 that have contributed to the enhancement of migration after skin injury [12,35,36]. Taken together, K16 appears to be an important protein in cell migration, which is considered a major event in the metastatic cascade.

CTCs in breast cancer are known to be an indicator for poor outcome [37]. However, the potential of individual CTCs to initiate metastasis is heterogeneous [38]. Here, we identified three CTC subsets that exhibited different expression patterns of keratins (e.g., K16+/C11−, K16−/C11+, K16+/C11+) during tumor progression in patients. Previous studies have shown that phenotypic and genetic heterogeneity among CTCs are associated with a higher risk for metastasis [39,40,41]. In our preliminary analysis, K16 expression was associated with higher tumor aggressiveness, and it was mainly associated with epithelial–mesenchymal plasticity. Indeed, a significant relationship has been reported in several studies between the CTCs with high epithelial–mesenchymal plasticity and poor clinical outcomes in different cancer entities [5,42,43,44,45,46]. These CTCs have dynamic plasticity to adapt to the selective microenvironment during their dissemination in distant organs [5,43]. Although our pilot study has only analyzed a small collection of blood samples, a significantly shorter relapse-free survival was observed among patients diagnosed with K16-positive CTCs, which is consistent with our previous in silico analyses [3]. Although additional studies with larger cohorts and the analyses of more confounding clinical factors are required to elucidate the true correlation between K16 expression among CTCs and relapse-free survival, this study can be considered hypothesis generating.

To our knowledge, this has been the first study to assess the role of K16 in metastatic breast cancer. Our results suggest that K16 may represent a novel metastasis-associated protein that acts as a positive regulator of cell motility and promoter of EMT regulator genes in breast cancer. This is an important issue for future research to analyze the exact function of K16 in tumor dissemination and metastasis development by assessing K16 status in disseminating tumor cells (DTCs) and CTCs. Additionally, understanding the biology of K16 would provide valuable mechanistic insights translating into a novel therapeutic opportunity.

## 5. Conclusions

Our results provide new insights into K16, representing a novel metastasis-associated protein for breast cancer, enhancing cell motility and promoting EMT. K16 expression in CTCs contributes to shorter relapse-free survival in metastatic breast cancer patients. This has been the first report indicating that K16 might be a metastasis-promoting gene in breast cancer.

## Figures and Tables

**Figure 1 cancers-13-03869-f001:**
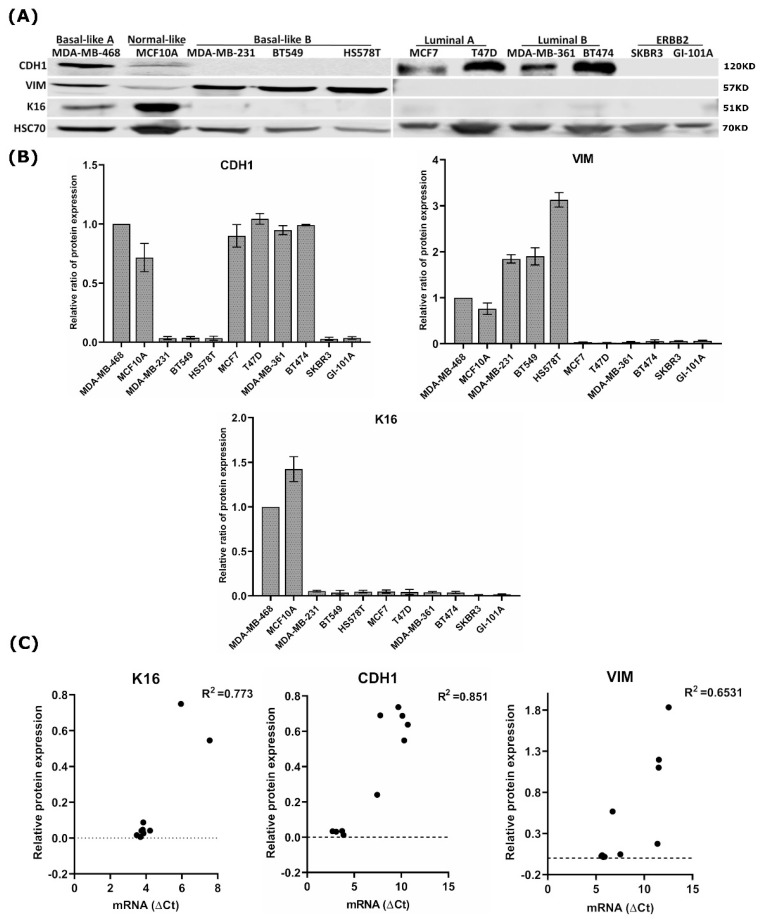
Analysis of the K16 expression in breast cancer cell lines. (**A**) Expression of K16, adherent junction protein CDH1, and mesenchymal intermediate filament VIM in a panel of human breast cancer cell lines using western blot. (**B**) The relative ratio of protein expression of K16, VIM, and CDH1; all quantified values are normalized to the expression level of HSC70 protein and then to MDA-MB-468 as a control (**C**) Correlation between mRNA and protein expression levels of K16, CDH1, and VIM in breast cancer cell lines.

**Figure 2 cancers-13-03869-f002:**
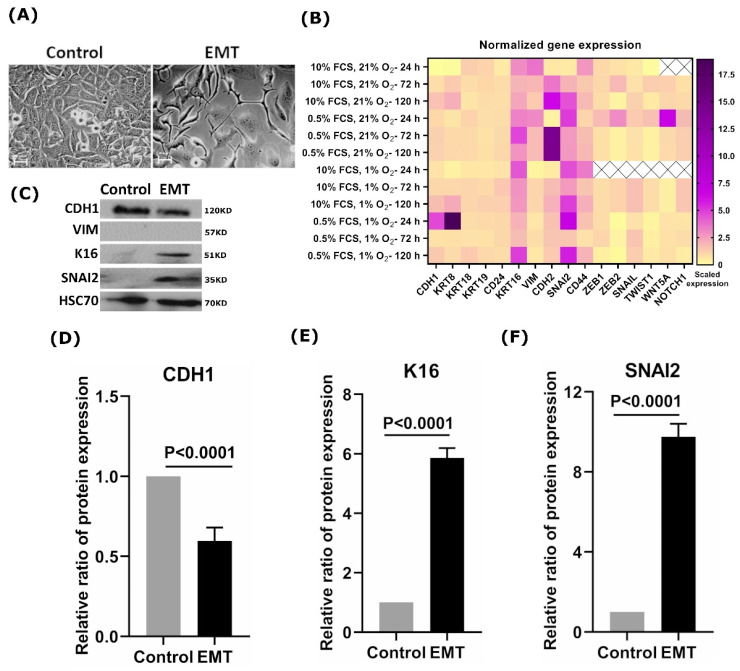
Analysis of the K16 expression in EMT-induced cells. (**A**) Microscopic images showed morphological changes in MCF7 cells during EMT induction; scale bar represents 100 μm. (**B**) Heat-map of EMT/MET specific genes expression in EMT-induced cells at different culture conditions and time points; relative mRNA expression values were normalized to *GAPDH* and subsequently displayed relative to gene expression as a fold change 2^−(∆∆CT)^. (**C**) Protein expression profile of K16, CDH1, VIM, and SNAI2 during EMT induction on MCF7 cells. HSC70 was used as a reference control. The relative ratio of protein expression of (**D**) CDH1; (**E**) K16; (**F**) SNAI2.

**Figure 3 cancers-13-03869-f003:**
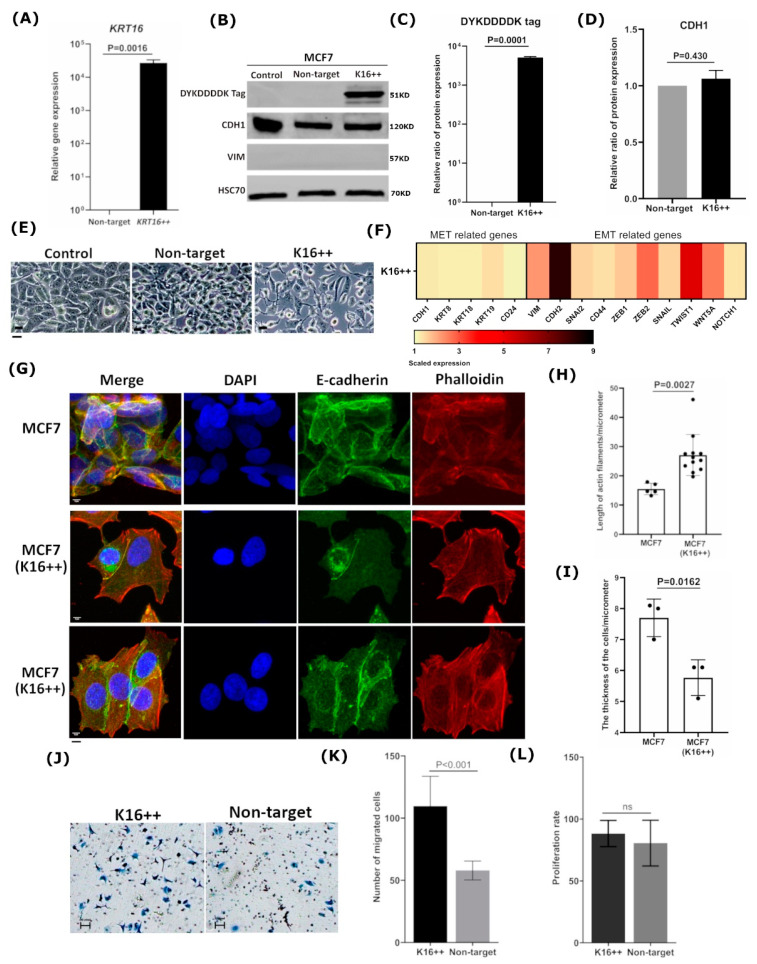
Overexpressing of K16 in MCF7 cells promotes a mesenchymal phenotype. (**A**) Relative mRNA expression was verified after inducing KRT16. (**B**) Immunoblot: DYKDDDDK tag was used to detect the expression of transfected K16 in MCF7 cells (K16++) compared to transfected empty vector (non-target) and untreated cells (control); CDH1 and VIM expression were investigated after 48h of treating MCF7 cells, loading control: HSC70. The relative ratio of protein expression of (**C**) DYKDDDDK tag and (**D**) CDH1. (**E**) Cellular morphological changes in MCF7 cells after inducing the expression of K16 protein (K16++) were observed under a normal microscope compared to transfected non-target and untreated cells (control); scale bars 20 μm. (**F**) Heat-map of EMT/MET-specific genes was investigated after KRT16 enhancement; mRNA expression values were normalized to GAPDH and subsequently displayed relative to gene expression as a fold change 2^−(∆∆CT)^. (**G**) Immunocytochemistry staining of actin filaments by phalloidin (red), intercellular adhesion by E-cadherin (green), and nucleus by DAPI (blue) to visualize the morphological changes in MCF7 cells that induced K16; scale bar represents 5 μm. Differences between MCF7 cells and MCF7 cells with induced K16 in (**H**) the length of actin filaments and (**I**) the thickness of the cells. (**J**) Microscopy images of migrated cells by Boyden Chamber to analyze the motility of MCF7 cells overexpressing K16 relative to the non-target control; the cells were seeded on transwell chambers and incubated for 24 h; scale bar represents 100 μm. (**K**) The migrated cells were counted after staining cells with crystal violet, the data were generated as the mean ± SD., *n* = 3. (**L**) Proliferation assay: cell proliferation rates of K16++ and non-target control cells were determined using Trypan Blue Dye, and the cells were counted by Vi-CELL Counter device; the data are expressed as the mean ± SD.; *n* = 3. The *p*-values were calculated with Welch’s two sample *t*-test (*p* < 0.05).

**Figure 4 cancers-13-03869-f004:**
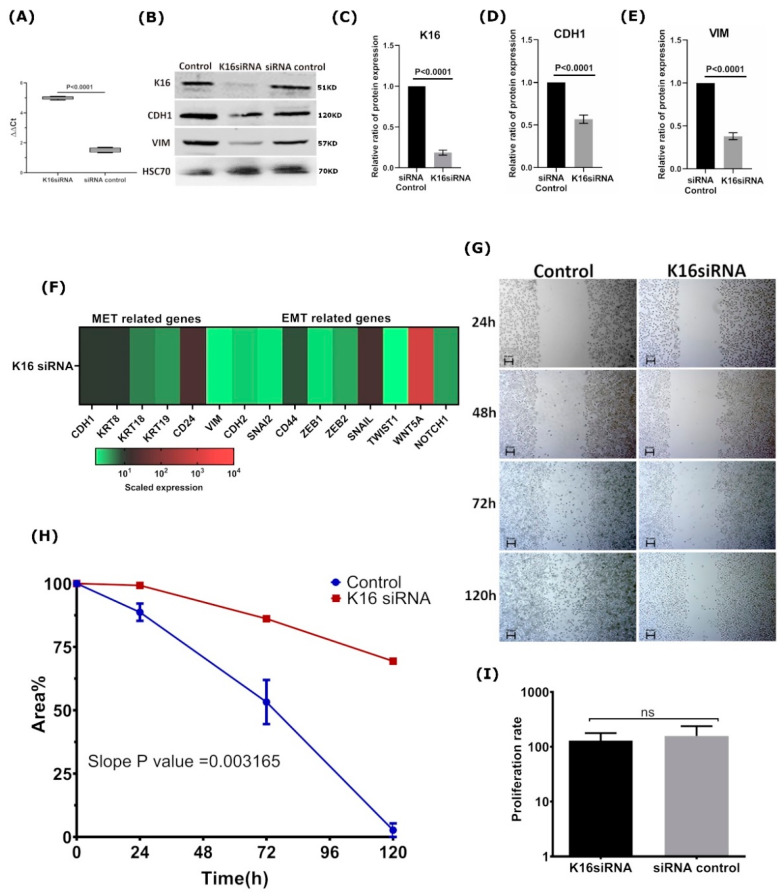
*KRT16* knockdown in MDA-MB-468 impaired cell migration (**A**) mRNA expression level was verified after *KRT16* knockdown compared to the non-target (siRNA control). (**B**) Immunoblot to verify the expression levels of CDH1 and VIM in *KRT16* depleted cells. The relative ratio of protein expression of (**C**) K16; (**D**) CDH1; and (**E**) VIM. (**F**) Relative mRNA expression of EMT/MET-specific genes was investigated after *KRT16* depletion; mRNA expression values were normalized to *GAPDH* and subsequently displayed relative to gene expression as a fold change 2^−(∆∆CT)^. (**G**) Microscopy images of migrated cells by wound-healing assay were performed on *KRT16* depleted cells (*KRT16* siRNA) compared to control cells (siRNA control) at different time points of 0 h, 24 h, 72 h, and 120 h; *n* = 2, scale bar represents 100 μm. (**H**) In migration, linear regression analysis was performed for the velocity slope of *KRT16*-siRNA transfected cells and untreated cells (control). (**I**) Cell proliferation rates of *KRT16*-siRNA transfected and siRNA control cells were determined using Trypan Blue Dye, and the cells were counted by a Vi-CELL Counter device, the data are expressed as the mean ± SD.; *n* = 3. The *p*-values were calculated by the Wilcoxon rank-sum test (*p* < 0.05).

**Figure 5 cancers-13-03869-f005:**
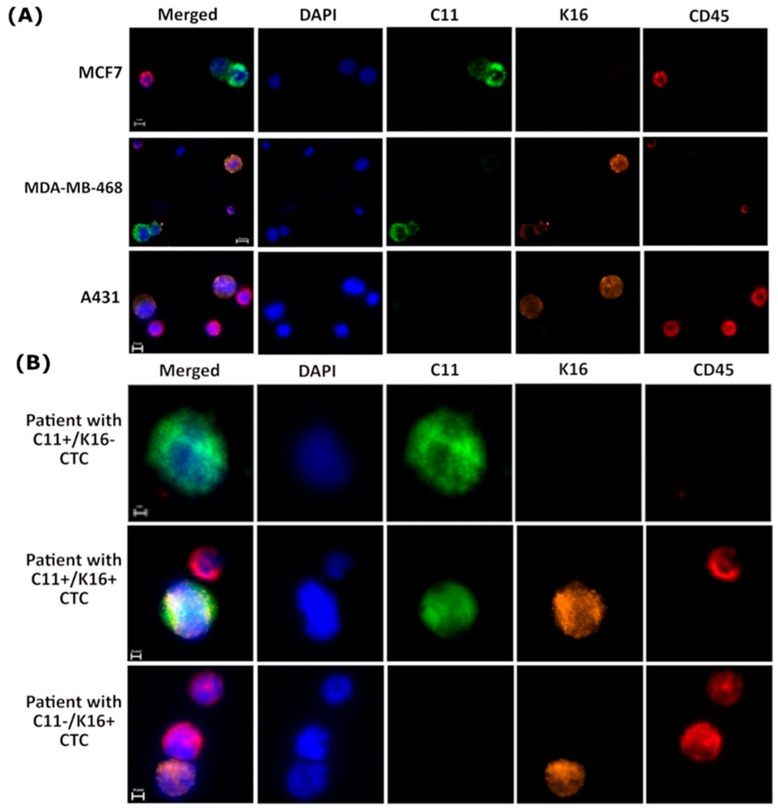
K16 immunocytochemistry staining (**A**) Establishment K16 immunocytochemistry staining; cells were stained by DAPI (blue), CD45 (AF647; red), C11 (AF488; green), and K16 (AF546; orange); scale bar represents 10 μm. (**B**) Detection of CTCs in metastatic breast cancer patients (*n* = 20); CTCs cells (DAPI+, C11+, K16+, and CD45−) were differentiable from the WBCs (DAPI+ and CD45−); scale bar represents 10 μm.

**Figure 6 cancers-13-03869-f006:**
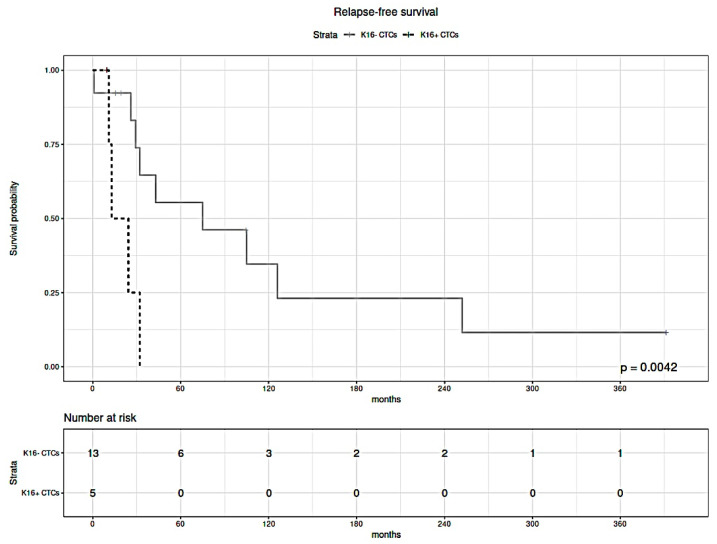
Kaplan–Meier estimates of the relapse-free survival of metastatic breast cancer patients with K16+ CTCs (black dashed line) and K16− CTCs (solid gray line). Statistical significance was determined by a log-rank test. Shorter relapse-free survival correlates with the presence of K16+ CTCs in the blood (*p* = 0.0042).

**Figure 7 cancers-13-03869-f007:**
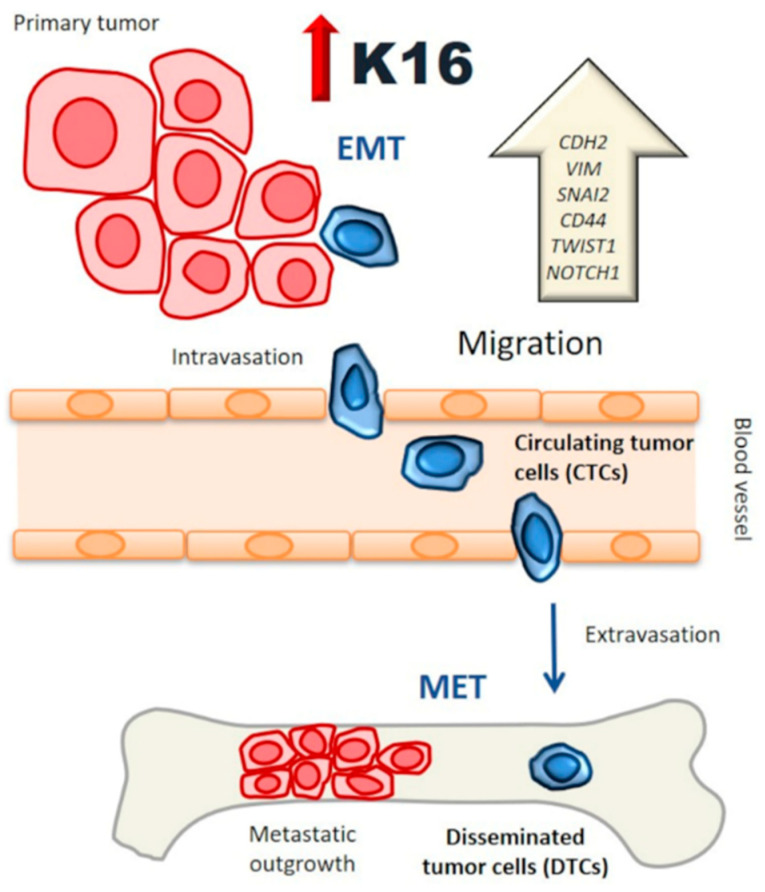
K16 functional model: K16 is involved in a functional transition of polarized epithelial cells to motile by promoting EMT regulator genes while preserving the epithelial phenotype, enhancing the migration of tumor cells to access the bloodstream and survive as a circulating tumor cell (CTCs). These tumor cells have a semi-mesenchymal phenotype with a high capacity to generate metastasis.

**Table 1 cancers-13-03869-t001:** Specific primers used in RT-qPCR.

Gene	Forward Primer (5’-3′)	Reverse Primer (5’-3′)	Product Length
**E-cadherin (*CDH1*)**	CGAGAGCTACACGTTCACGG	GGGTGTCGAGGGAAAAATAGG	119 bp
**N-cadherin (*CDH2*)**	TGCGGTACAGTGTAACTGGG	GAAACCGGGCTATCTGCTCG	123 bp
**Vimentin (*VIM*)**	GACGCCATCAACACCGAGTT	CTTTGTCGTTGGTTAGCTGGT	238 bp
***SNAI2* (*SLUG*)**	TGTGACAAGGAATATGTGAGCC	TGAGCCCTCAGATTTGACCTG	203 bp
***SNAI1***	ACTGCAACAAGGAATACCTCAG	GCACTGGTACTTCTTGACATCTG	242 bp
***ZEB1***	GATGATGAATGCGAGTCAGATGC	ACAGCAGTGTCTTGTTGTTGT	86 bp
***ZEB2***	GGAGACGAGTCCAGCTAGTGT	CCACTCCACCCTCCCTTATTTC	107 bp
***TWIST1***	AAGGCATCACTATGGACTTTCTCT	GCCAGTTTGATCCCAGTATTTT	96 bp
***WNT5A***	ATTCTTGGTGGTCGCTAGGTA	CGCCTTCTCCGATGTACTGC	159 bp
***NOTCH1***	GAGGCGTGGCAGACTATGC	CTTGTACTCCGTCAGCGTGA	140 bp
***KRT8***	CAGAAGTCCTACAAGGTGTCCA	CTCTGGTTGACCGTAACTGCG	194 bp
***KRT18***	GCTCAGATCTTCGCAAATACTGT	CTTCCTCTTCGTGGTTCTTCTTC	250 bp
***KRT19***	ACCAAGTTTGAGACGGAACAG	CCCTCAGCGTACTGATTTCCT	181 bp
***KRT16***	GACCGGCGGAGATGTGAAC	CTGCTCGTACTGGTCACGC	91 bp
***CD24***	CTCCTACCCACGCAGATTTATTC	AGAGTGAGACCACGAAGAGAC	166 bp
***CD44***	CTGCCGCTTTGCAGGTGTA	CATTGTGGGCAAGGTGCTATT	109 bp
***GAPDH***	GGAGCGAGATCCCTCCAAAAT	GGCTGTTGTCATACTTCTCATGG	197 bp

**Table 2 cancers-13-03869-t002:** Cox proportional hazard ratios. Estimated coefficients of relapse-free survival on breast cancer subjects. Calculated are the corresponding hazard ratio (HR), 95% confidence interval (CI) of the hazard ratio, and *p*-value in uni- and multivariable Cox proportional hazard analysis for the presence of K16-positive CTCs, estrogen receptor (ER) status, progesterone receptor (PR) status, ERBB2 status, and T-stage (reference T1-2).

	Univariable Analysis				Multivariable Analysis			
Covariate	Coefficient (bi)	HR [exp(bi)]	HR 95% CI	*p*-Value	Coefficient (bi)	HR [exp(bi)]	HR 95% CI	*p*-Value
**K16+ CTCs**	2.061	7.856	(1.677, 36.81)	0.0089	4.864	129.6	(1.6, 1.120)	0.0287
**Age**	−0.072	0.930	(0.867, 0.998)	0.0432	−0.217	0.804	(0.627, 1.032)	0.0869
**ER+**	−0.638	0.528	(0.166, 1.680)	0.280	−5.909	0.003	(0.0, 0.951)	0.0481
**PR+**	−0.150	0.861	(0.275, 2.693)	0.797	3.857	47.3	(0.108, 2.670)	0.2136
**ERBB2+**	0.078	1.081	(0.283, 4.123)	0.910	−0.435	0.647	(0.030, 14.1)	0.7820
**T3-4**	0.272	1.312	(0.286, 6.420)	0.737	0.117	1.352	(0.080, 15.9)	0.9308

CTCs = circulating tumor cells, ER = estrogen receptor, PR = progesterone receptor, ERBB2 = Erb-B2 Receptor Tyrosine Kinase 2, HR = hazard ratio.

## Data Availability

The datasets generated during and/or analyzed during the current study are available from the corresponding author on request.

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
