# Peer review of "Emerging Insights into Keratin 16 Expression during Metastatic Progression of Breast Cancer"

_cancers, 2021, doi:10.3390/cancers13153869_

Round 1

Reviewer 1 Report

The authors have properly addressed my concerns.

Especially the addition that this study can be considered hypothesis generating.

Author Response

Point-by-point reply to the comments raised by the reviewers with respect to the manuscript " Emerging insights into keratin 16 expression during metastatic progression of breast cancer" by Maha Elazezy et al.

We would like to thank the reviewers for their time and effort to provide their reviews. We have carefully revised the manuscript according to the suggestions provided.

Reviewer 1 - Comments and Suggestions for Authors.

The authors have properly addressed my concerns.

Especially the addition that this study can be considered hypothesis generating.

We thank the reviewer for their valuable comments.

Reviewer 2 Report

Although most of the original concerns have been addressed, the revised manuscript contains a new set of concerns. Although they are minor, these following issues should be addressed:

  • In line 648, this reviewer is assuming that the authors are referring to the transfection of K16 in MCF7 cells. If so, the authors should specify by saying transfection of K16 in MCF7 cells Instead of “K16 stimulation.”
  • The word “Although” in line 646 seems out of context.
  • Lines 653-658, the sentence is too long and confusing. In it, the authors do acknowledge that CDH1 level was reduced in K16-depleted cells (MDA-MB-468?), but also claim that the expression of epithelial markers are stably maintained. As CDH1 is a key epithelial marker, this sentence needs to be rephrased.
  • There are no figure legends for supplementary figures.
  • Figures S2 and S3 are not adequately labeled.

Author Response

Point-by-point reply to the comments raised by the reviewers with respect to the manuscript " Emerging insights into keratin 16 expression during metastatic progression of breast cancer" by Maha Elazezy et al.

We would like to thank the reviewers for their time and effort to provide their reviews. We have carefully revised the manuscript according to the suggestions provided.

All modifications in the revised manuscript are marked up using the “Track Changes” function.

Reviewer 2- Comments and Suggestions for Authors.

Although most of the original concerns have been addressed, the revised manuscript contains a new set of concerns. Although they are minor, these following issues should be addressed:

  • In line 648, this reviewer is assuming that the authors are referring to the transfection of K16 in MCF7 cells. If so, the authors should specify by saying transfection of K16 in MCF7 cells Instead of “K16 stimulation.”

In response to the reviewer's comment, we have changed the Discussion of the manuscript, on Page 15 as follows:

The relative gene and protein expression of epithelial markers such as CDH1 were not changed upon transfection of K16 in MCF7 cells, E-cadherin was relocalized from the cell membrane to the cytoplasm and thereby compromising cell-cell adhesion, as well as acting possibly as transcriptional gene regulator [4, 31].”

  • The word “Although” in line 646 seems out of context.

In response to the reviewer's comment, we have made the following change to the Discussion of the manuscript, on Page 15:

The relative gene and protein expression of epithelial markers such as CDH1 were not changed up-on transfection of K16 in MCF7 cells, E-cadherin was relocalized from the cell membrane to the cytoplasm and thereby compromising cell-cell adhesion, as well as acting possibly as transcriptional gene regulator [4, 31].”

  • Lines 653-658, the sentence is too long and confusing. In it, the authors do acknowledge that CDH1 level was reduced in K16-depleted cells (MDA-MB-468?), but also claim that the expression of epithelial markers are stably maintained. As CDH1 is a key epithelial marker, this sentence needs to be rephrased.

In response to the reviewer's comment, we have made the following changes to be more clear in the Discussion of the manuscript, on Page 15:

“Furthermore, the knockdown of KRT16 also led to the downregulation of mesenchymal-associated genes, but not epithelial-related genes. Whereas the expression level of CDH1 protein was reduced in K16-depleted cells, our results suggest that expression of K16 is correlated with the regulation of the mesenchymal-regulatory genes of EMT as well as the expression of CDH1 as a crucial epithelial marker.”

  • There are no figure legends for supplementary figures.

In order to respond to the reviewer’s comments, legends of the supplementary data files are now located in the section of Supplementary Materials of the manuscript, on page 17:

“Supplementary Materials: The following are available online at www.mdpi.com/xxx/s1, Figure S1: in silico analysis, KRT expression in breast cancer cell lines, the red color represents high relative gene expression, and the blue color represents low relative gene expression., Figure S2: Transfection efficiency of KRT16, immunocytochemistry staining of transfected cells compared to non-target vector, the cells were stained by DYKDDDDK Tag (AF488; green) and DAPI (blue), scale bar represents 20 μm., Figure S3: Immunofluorescence staining of K16 (AF488; green) and DAPI (blue), scale bar represents 10 μm.,  Figure S4: Immunocytochemistry staining to visualize the ac-tin filaments by phalloidin (red), intercellular adhesion by E-cadherin (green), and nucleus by DAPI (blue) in MDA-MB-468, MCF7, and MCF7 cells that induced K16, scale bar represents 5 μm., Table S1: Transfection efficiency of KRT16., Table S2: Number of detected CTCs with patient's characteristics., Video S1: 3D imaging view of the actin filaments rearrangement (red), E-cadherin localization (green), and DAPI (blue) in MCF7 cells that induced K16.”

  • Figures S2 and S3 are not adequately labeled.

In response to the reviewer's comment, we adapted Figure S2 and S3 accordingly.

This manuscript is a resubmission of an earlier submission. The following is a list of the peer review reports and author responses from that submission.

Round 1

Reviewer 1 Report

The authors present a study that investigates the keratin 16 expression on CTC both in vitro and in vivo.

By performing RT-qPCR, western blot, and immunocytochemistry they adequate showed the expression of keratin 16 in breast cancer cell lines and showed the association of keratin 16 with intermediate mesenchymal phenotype, regulatory effect on EMT and cell motility in these cell lines.

However their conclusion on keratin 16 expression on small cohort of patient CTCs and association with survival are inconsistent and not correct, based on the reported results.

  • Which 5 patients from table S2 represents the K16+ CTCs group in the KM plot and survival analysis?
  • Why this K16+ group has only 5 patients and the K16- group 14 patients when authors report that more than 80% of CTC are K16 positive and almost all patients have K16+ CTCs and/or K16+,C11+ CTCs in table S2?   
  • The conclusion that presence of K16-positve CTCs was correlated with a shorter worse is not correct, based on the results in Table S2.
  • Patient 5 and 8 are reported without any CTCs in table S2, so the number of patients with detectable CTCs is 18/20 and not 19/20 as reported, please verify.
  • Please report which data was used to calculate the median follow-up time of 43.0 months for all patients, this number cannot be derived from the reported follow-up numbers in table S2.
  • Same for the 5 K16+ patients with a shorter survival median of 18.6, no patients is reported with a follow-up of 18,6 months.

Reviewer 2 Report

The manuscript discusses the potential role of keratin 16 as a promoter of metastasis in breast cancers. The manuscript does a nice job with inclusions of in silico analysis, in vitro experiments, and data from patient samples. The major issue with the manuscript is the limited patient sample size used. In addition, there are several issues that need to be addressed before the manuscript can be considered for publication.

  • Dependence of CDH1 expression on K16 is quite interesting. One might expect levels of mesenchymal markers to increase in K16 expressing cells that exhibit spindle-shape morphology and increased migration. However, CDH1 expression was also dependent on K16. The authors should elaborate on why level of epithelial marker CDH1 becomes upregulated in mesenchymal cells where cell-cell contacts are compromised and cell migration is increased.
  • On the topic of K16, the authors claimed that CDH1 mRNA expression increases upon K16 expression but in Fig 3B, 3D and 3F, no changes in CDH1 levels were observed.
  • Although there are increases in mRNAs of mesenchymal markers including VIM (Fig 3F), VIM protein level remained unchanged (Fig 3B). In light of this, the authors should test protein levels of CDH2 or TWST1 which showed greater increase upon K16 overexpression.
  • For Fig 3E, non-target cells seem to be much larger than control or K16++ cells. If this is true, the authors should quantitate the finding. Otherwise, more representative image should be chosen. Also, K16++ cells are mostly rounded which could be dying cells. Again, the authors should either quantitate the observation or choose more representative image.
  • The authors should perform K16 staining in K16++ cells to examine their filament status. This, together with K8 staining can also address the intriguing possibility of the presence of monomeric K16 filaments.
  • Since MDA-MB-468 cells express other keratins beside K16 (Fig S1), the authors should test the expression of closely related keratins to ensure that K16 siRNA specifically targeted K16.
  • For Fig 1B, it’s not at all clear how the graph was generated and values were normalized. For each of K16, VIM and CDH1, one would expect a control cell line where the expression was set to one. Also, in GI-101A cell line, there’s no CDH1 expression based on the WB data but the graph seems to indicate that there’s a good amount of CDH1 expressed in GI-101A cells.

Minor issues:

  • Fig S1 would be more useful if the authors show which subtypes (e.g. TNBC) cell lines belong to.
  • For some figures, such as Fig 2D-F, there are no error bars. The authors should present error bars and p-valued for all graphs.
  • Line 409 – not only mesenchymal but also epithelial as well
  • Presumably, the authors are referring to a blend of epithelial (high CDH1) and mesenchymal (high VIM) status when using the term “a hybrid phenotype” in lines 526-535. However, this is not obvious from the paragraph and should be clarified.
  • Refs 12 & 31 are same.

Reviewer 3 Report

The aim fo this article was to investigate the biological role and the clinical relevance of K16 in metastatic breast cancer. Article is concisely written; conclusions are supported by the data, however, there are some issues that should be resolved. 

Major points - I suggest to better characterize metastatic breast cancer patients, their biological subtype, and relationship between blood sampling and treatment status as these could have major impact on study results and data interpretation. Due to limited sample size (20 patients only) there could be many clinical and biological confounders that effect study results.  - Authors should be better described positivity of KRT16+ CTC, e.g. if there was cut-off for this positivity, or number of KRT16+ cells that should be present to categorize patient as KRT16+CTC. I suggest, that clinical and biological difference between patient with 1 of 10 CTC KRT16+ vs. 9 of 10.